# Using big sequencing data to identify chronic SARS-Coronavirus-2 infections

**Sheri Harari**[1,2,4], **Danielle Miller**[1,2,4], **Shay Fleishon**[3], **David Burstein** [1,2] & **Adi Stern** [1,2] ✉

The evolution of SARS-Coronavirus-2 (SARS-CoV-2) has been characterized by the periodic emergence of highly divergent variants. One leading hypothesis suggests these variants may have emerged during chronic infections of immunocompromised individuals, but limited data from these cases hinders comprehensive analyses. Here, we harnessed millions of SARS-CoV-2 genomes to identify potential chronic infections and used language models (LM) to infer chronic-associated mutations. First, we mined the SARS-CoV-2 phylogeny and identified chronic-like clades with identical metadata (location, age, and sex) spanning over 21 days, suggesting a prolonged infection. We inferred 271 chronic-like clades, which exhibited characteristics similar to confirmed chronic infections. Chronic-associated mutations were often high-fitness immune-evasive mutations located in the spike receptor-binding domain (RBD), yet a minority were unique to chronic infections and absent in global settings. The probability of observing high-fitness RBD mutations was 10-20 times higher in chronic infections than in global transmission chains. The majority of RBD mutations in BA.1/BA.2 chronic-like clades bore predictive value, i.e., went on to display global success. Finally, we used our LM to infer hundreds of additional chronic-like clades in the absence of metadata. Our approach allows mining extensive sequencing data and providing insights into future evolutionary patterns of SARS-CoV-2.

The evolution of SARS-Coronavirus-2 (SARS-COV-2) has been punctuated by the periodic emergence of variants that are highly genetically divergent compared to the circulating variants at the time of their emergence. Some of these variants were found to be more transmissible than their predecessor variants, leading to patterns of rapid displacement of one variant by another. Specific variants of concern (VOCs) were designated by the World Health Organization (WHO) when there was concrete evidence that a variant posed an increased risk to public health. The nomenclature of SARS-COV-2 variants has been challenging[1]; here we refer to VOCs mainly based on the Greek letters assigned by PANGO[2] and WHO[1]. When more precise nomenclature is necessary, we rely on the classification of Nextstrain clades[3].

As of the end of 2020, a series of VOCs created patterns of global displacement, namely Alpha, Delta, Omicron BA.1, Omicron BA.2, and lastly Omicron BA.5. More recently, milder global changes have been occurring, as at the time of writing there are multiple variants co-circulating globally, and these variants are usually not dramatically divergent compared to their predecessors.

Several hypotheses have been raised regarding the origin of highly divergent SARS-COV-2 variants, namely, undetected circulation in regions where sequencing is sparse, a zoonotic origin, and emergence in chronically infected individuals (also called persistent or prolonged infections). Notably, it is very hard to definitively prove any of these hypotheses. The latter hypothesis has been gaining mounting

[1]The Shmunis School of Biomedicine and Cancer Research, Tel Aviv University, Tel Aviv, Israel. [2]Edmond J. Safra Center for Bioinformatics, Tel Aviv University, Tel Aviv, Israel. [3]Israeli Health Intelligence Agency, Public Health Division, Ministry of Health, Jerusalem, Israel. [4]These authors contributed equally: Sheri Harari, Danielle Miller. ✉e-mail: sternadi@tauex.tau.ac.il

support[4–13], mainly due to the fact that different combinations of VOC lineage-defining mutations, particularly in the spike gene, are observed across chronic infections[4,7,9,14].

Chronic infections are herein defined as infections where there is evidence of actively replicating virus for more than 21 days, and often such infections may last months or even more than a year[14]. To date, chronic infections were found predominantly in immunocompromised individuals suffering from one of four categories: hematologic cancer, AIDS, transplant patients or autoimmune patients[15–19]. Chronic infections should not be confused with long COVID where symptoms persist but not necessarily active viral replication.

Notably, epidemiologic modelling of chronic infections and their impact on global circulation has led to some contradictory results. Early models suggested that chronic infections would not likely lead to variants that spread in the population, mainly due to the assumption that chronically infected individuals tend to be isolated and rare[20]. Isolation of patients is true in some settings but not in others, and while the assumption of rarity is likely true, during the current pandemic, rare events translate to large numbers. Importantly, more recent modelling papers focusing on SARS-CoV-2, do support the possibility that chronic infections may harbor variants that are infectious and contribute to global spread[21–25].

Despite the increased interest in chronic SARS-COV-2 infections, our understanding of the dynamics of these infections and how they correlate with global evolutionary patterns is limited and mostly contingent on isolated case reports. Our and other previous meta-analyses[14,26] were mostly limited to the pre-VOC era and were based on a small number of cases (but see[27]). We posit here that the ever-growing database of millions of SARS-COV-2 genomes likely harbors many sequences derived from chronic infections. We adopt a two-step approach: first, utilizing phylogeny and sequencing metadata to identify potential clades, and then employing language models to analyze these clades and use the information to infer clades in the absence of reliable metadata.

Notably, sequences are usually not explicitly identified as originating from chronically infected individuals or from the same person. We reasoned that most often, sequences derived from chronically infected individuals will display as monophyletic clades, i.e., all sequences derived from the ancestral node of the clade are from the same individual[16,18,19] and share metadata (age/sex/location, and range of dates). This reflects the following assumptions: (i) sometimes, chronically infected individuals will be serially sampled and sequenced, (ii) sequences derived from the same individual will be very similar, and (iii) most chronically infected patients do not create onward transmission chains[14,27] (otherwise, the clade would not be monophyletic). We accordingly mined a global phylogeny of over 11 million sequences and inferred 271 inferred chronic-like monophyletic clades and a set of control clades derived from transmission chains. By comparing chronic-like clades to controls, we were able to obtain insights regarding the different evolutionary dynamics of chronic versus acute infections.

Analysis of data at the scale described herein poses many challenges, including missing data and computational limitations. Recently, language models have gained widespread popularity due to their ability to tackle such large volumes of data and have impacted research in diverse areas of biological sciences, including virology[28–30]. To this end, we used language models to effectively capture the distinct mutational patterns exhibited by chronic-like clades in comparison to control clades. This enabled us to identify mutations that exhibit a stronger association with chronic-like clades. Finally, we used this model to infer chronic infections, in the absence of metadata. Our analysis allowed us to demonstrate that in-depth mining of the huge volume of SARS-CoV-2 sequences is highly informative and that chronic-like clades may sometimes hold predictive value for inferring future evolution.

## Results

We began by mining a phylogeny of over 11.7 million sequences of SARS-COV-2 for monophyletic clades where all sequences share homogenous metadata (defined here as location, age, and sex) and where the sampling dates of the sequences span more than 21 days, resulting in 271 chronic-like clades (also denoted as cases). Notably, only ~25% of sequences ($n = 2,929,351$) bore valid metadata regarding sex/age/dates and passed the sequence quality control we applied (Methods). In addition, we constructed a set of positive and negative control clades. The set of positive clades was composed of thirty-two bona fide chronic infections derived from published case reports for which sequencing, metadata and clinical information were available (Methods). The set of negative control clades was generated by sampling 15,163 monophyletic clades with mixed metadata (Methods) (Fig. S1), to ensure that these clades most likely represent transmission chains from acute infections. In the analyses below, when comparing between cases and controls, we performed stratified sampling from this set of controls to maintain identical sample sizes of $n = 271$, and to maintain similar distributions of clade sizes and background variants between cases and controls (Methods).

The 271 chronic-like clades were distributed across all the major variants detected till September 2022 (Fig. 1a, Table S1) and originated from a wide range of countries, with a small bias towards Europe (Table S2). Average clade size was 4 sequences, and the average time span was 55 days (Fig. S1). Of the 271 clades 182 (67.15 %) were 100% concordant in metadata, and the remaining 89 (32.85%) were concordant in over 75% of the metadata (Methods). We first set out to test whether we could find evolutionary signatures that would suggest that the set of chronic-like clades found herein are enriched for valid chronic infections.

We compared basic demographic features of cases and controls, namely age and sex. We found that on average chronic-like clades were characterized by older age ($p < 0.001$, $t$-test), and a higher proportion of males ($p < 10^{-4}$, permutation test; Methods), as compared to the control clades (Fig. 1b). This trend highly resembled the bona fide chronic infections. This is in line with a higher tendency for older males to suffer from hematologic cancers, which represented the majority of chronic infections in our previous study[14]. Additionally, these individuals are more likely to present with COVID-19 deteriorations or hospitalizations[31,32], possibly leading to a higher chance of being sequentially sampled.

We addressed the possibility that our approach may capture outbreaks that occurred in particular settings, such as in a school, hospital or care home, which by chance may lead to concordant monophyletic clades with same age/sex/location. First, we searched for such clades during the early stages of the pandemic, by focusing on sequences sampled till March 31, 2020. We surmised that the probability of repeatedly sampling an immunocompromised individual at this early stage was very low, and moreover, ours and other studies from this time-period intentionally sampled unrelated individuals[33]. Thus, this would allow us to obtain a rough estimate of the chance of observing a concordant clade. Out of 27,627 sequences with valid metadata and after quality control, zero formed a concordant clade as defined herein (Methods); this is as opposed to 1129 sequences forming our 271 chronic-like clades, out of 2,929,351 sequences post quality control (i.e., sequence detection rate of $3.9 \times 10^{-4}$). Of note, this is a small difference, and sampling strategies varied between early and mid/late stages of the pandemic, suggesting caution when interpreting this result. A second sanity check that we performed, was to examine sequences in the clades that neighbor our chronic-like clades. If chronic-like clades were mostly derived from an outbreak in a high-age setting, then neighboring clades would probably display higher ages as well. However, our results show that neighboring clades show the same average age as controls, which is lower than that of chronic-like

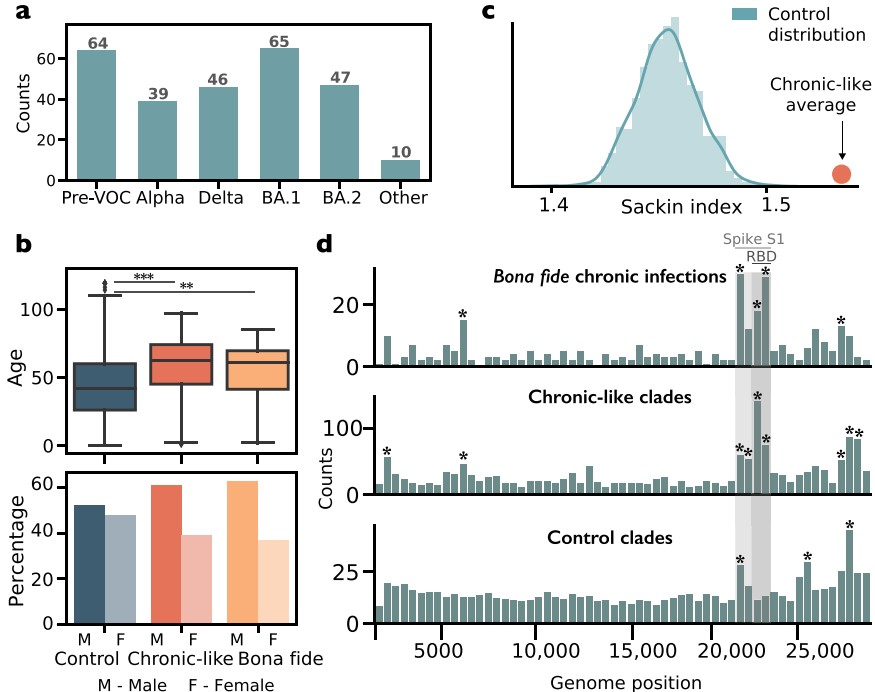

**Fig. 1 | Characteristics of chronic-like clades compared to control clades and bona fide chronic infections. a** The number of chronic-like clades stratified by background variant. Pre-VOC refers to any variant that was dominant before the emergence of Alpha. **b** Distribution of age and percentage of male/female shown for control clades ($n = 15,163$), chronic-like clades ($n = 271$) and bona fide chronic infections ($n = 32$). The central line of box defines the median value. The box itself encompasses the interquartile range (IQR), with the lower and upper edges indicating the 25th and 75th percentiles, respectively. On average chronic-like clades were characterized by older age ($p < 0.001$, two-tailed $t$-test), and a higher

proportion of males ($p < 10^{-4}$, permutation test). **c** The distribution of average Sackin index values over 10,000 repeated stratified sampling of $n = 271$ control clades, with the orange circle representing the average Sackin index of the chronic-like clades. **d** Distributions of substitutions along the SARS-CoV-2 genome observed across all bona fide chronic infections, chronic-like clades, and control clades. Substitutions are counted in bins of 500 nucleotides. Of note, for clarity the $y$-axis is not shared among the groups shown. Asterisks mark bins significantly enriched for more substitutions using a one-tailed binominal test, after correction for multiple testing ($p < 0.0001$, following false discovery rate (FDR) correction).

clades (Fig. S2). Overall, this suggests that our chronic-like clades are enriched for samples from the same chronically infected individual.

Next, we compared the distribution of substitutions found in the sets of bona fide chronic infections, chronic-like clades, and control clades, in bins of 500 bases along the genome (Fig. 1d). The bona fide chronic infections and chronic-like clades were both highly enriched for spike mutations in the S1 subunit of the spike protein. Chronic-like and bona fide share three of four enriched S1 bins, whereas controls share only one enriched S1 bin with the other two categories; the average number of S1 mutations per clade was significantly higher in bona fide and chronic-like clades than in control clades ($p < 5 \cdot 10^{-3}$, Mann–Whitney $U$ test; Supplementary text, Fig. S2). Previous research has shown repeated selection observed in globally distributed sequences at the S1 domain[34], most likely for antibody evasion and/or enhanced ACE2 binding. In contrast, the distribution of substitutions along the control clades was mostly uniform, with the exception of some enrichment in genes in the 3' region of the genome, found across all sets, and a small enrichment in the spike N-terminal domain. Indeed, it has been shown that many genes in the 3' region of the genome are under relaxed selection and tend to accumulate more mutations[35].

We went on to analyze the tree topology of the chronic-like clades. When comparing inter-host to intra-host evolution, we expect more ladder-like trees in the latter, reflecting adaptive evolution and step-wise accumulation of beneficial mutations over time[36]. On the other hand, acute infection transmission chains are expected to be char-acterized by superspreading events that lead to a star-like phylogeny. We focused on two features: the Sackin index of each sub-tree, and an entropy-based measure of each sub-tree (Methods), with higher values of both indices for ladder-like trees and lower values for star-like trees.

In line with this assumption, we found that on average our chronic-like clades bore a significantly higher Sackin index ($p < 10^{-4}$, permutation test, Fig. 1c) as well as a higher entropy value as compared to controls ($p < 10^{-8}$, Mann–Whitney test).

We present three examples of chronic-like clades (Fig. 2a–c) as well as an illustration of a control clade (Fig. 2d). As we describe herein, many chronic-like clades exhibit rapid evolution, especially in the spike gene. However, it is important to note that we also observed that in some chronic-like clades there was not necessarily dramatic evolution (e.g., Fig. 2c). This is in line with our previous report, where we observed that chronic infections varied in their inferred rate of adap-tive evolution[14].

We next went on to examine the evolutionary divergence in the chronic-like clades as compared to the controls (Methods). Notably, this is a challenging endeavor as the clades we analyzed span a very short time (ranging from 21 to 241 days), and sequences are not independent due to shared ancestry. Moreover, sequences towards the tips of the tree tend to be enriched with slightly deleterious mutations (so called incomplete purifying selection)[37]. Nevertheless, given the similar distributions of sizes and times across clades and controls (Fig. S1), we considered that averaging the regression slopes across a set of clades may allow us to compare trends between dif-ferent sets of clades. We thus employed a simple linear regression and regressed the number of mutations from the ancestral node, against the calendar day, for all sequences in each clade. The average slope of the regression lines was significantly higher in chronic-like clades as compared to controls ($p < 10^{-4}$, $t$-test), with average slopes corre-sponding to 16.63 and 12.56 mutations per year, respectively (Table S3). Reassuringly, the average slope in controls was in line with estimates of divergence obtained previously across large sets of

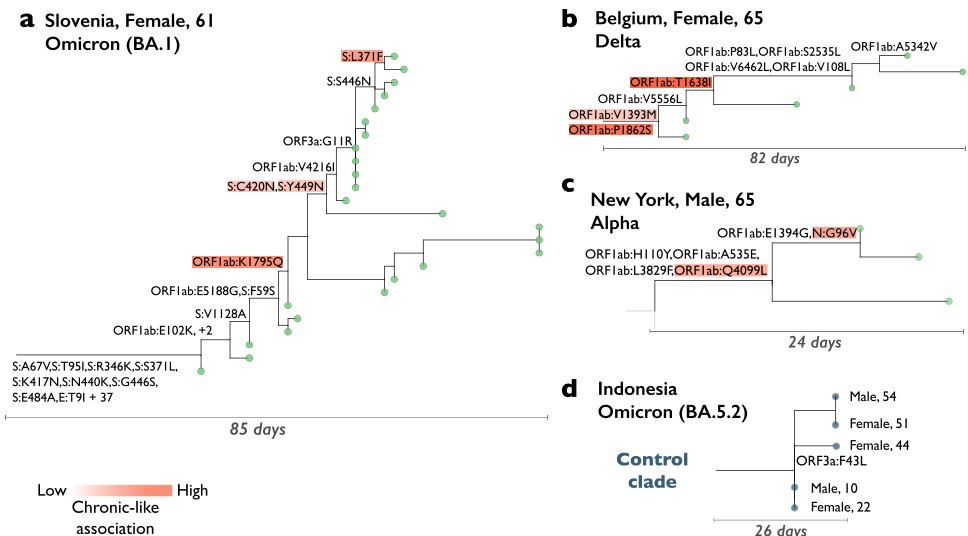

**Fig. 2 | Examples of chronic-like clades detected by our approach. a–c** Depict a chronic-like clades, where all sequences (tips) were presumably sampled from the same individual. **d** Illustrates a control clade. Only non-synonymous mutations are shown along the branches. In (**a–c**), mutations are color-coded based on their association with the group of chronic-like clades, with stronger shades of orange indicating a stronger association (see main text below on language models).

sequences from global transmission[35]. When focusing on synonymous mutations, we did not find any significant differences between the slopes of chronic-like versus control clades ($p = 0.37$, $t$-test). However, the average slope of non-synonymous mutations was significantly elevated, with a value of 13.8 in the chronic-like clades compared to 7.7 for control clades ($p < 10^{-10}$, $t$-test). This trend was exacerbated in the S1 subunit of the spike protein, where a X2.5-X4.5 higher average slope was found in chronic-like clades as compared to controls (Table S3). Overall, these results are highly suggestive of adaptive evolution occurring in the chronic-like clades.

We examined whether a specific VOC background was associated with different patterns of evolution in the chronic-like clades. An examination of the Sackin index did not reveal any differences among the chronic-like clades from different variants (Fig. S3). However, we noted that the regression slopes for non-synonymous mutations were lower in Delta as compared to BA.1 chronic-like clades ($p < 0.05$; one way ANOVA with Tukey's multiple comparisons test; Table S3). This could not be explained by differences in clade size or span of sampling dates (Fig. S1). We could not conclusively determine if a particular genomic region is driving the differences between Delta and BA.1 (Table S3) (see discussion).

We next set out to predict the sets of mutations that are most associated with chronic-like clades. This was quite challenging, since we noted quite a lot of noise in both the sequencing data and in the phylogenetic tree. In parallel to these problems, about 75% of sequences bore unreliable metadata, i.e., were labeled as "unknown" in at least one of the categories sex or age, and we reasoned that many of these sequences may have been derived from chronic infections. These challenges led us to adopt a deep learning approach that relied on language models (LMs) suitable to deal with large, noisy, and unlabeled data. Our goal was to design a classifier that would use the mutations most associated with chronic-like clades for predicting case from control sequences.

In our LM, "words" are mutations compared to the reference genome sequence, and thus a given SARS-CoV-2 genome is a sequence of words representing all its mutations as compared to a reference genome (Methods). We began by pre-training a BERT masked LM[38] and learned the conditional probabilities associated with observing a specific mutation within various mutational contexts. This allowed us to account for differences stemming from the background variants and timing (pre- or post-vaccination/convalescence). This is important

in part due to epistatic interactions among mutations, as observed repeatedly for SARS-CoV-2, particularly in Omicron[39–41].

Our LM model provided a numerical representation for each mutation. Projecting these representations onto a two-dimensional space allowed us to reassuringly verify that the model effectively separated sequences into their respective Nextstrain clades (Fig. S5, Methods). We next fine-tuned our pre-trained model to perform a classification task of distinguishing cases from controls. To test our classification process, we adopted a sequential cross-validation approach based on the time of emergence and the background variant (Methods). We found that the classifiers were successful in distinguishing between cases and controls with an average Area Under the Precision-Recall curve (AUPR) of 0.88, weighted by clade size and the number of mutations in the clade (Methods, Fig. S6).

To gain insight into the classifier's predictions for the cases, we employed LIME (Local Interpretable Model-agnostic Explanations)[42], a technique used to identify the "words" that have the greatest impact on model predictions and the reliability of each such inference (Methods). We focused on LIME scores higher than 0.05, which correspond to the highest 75% quantile. Several interesting observations emerged from this analysis. As expected, for most variants, non-synonymous spike mutations were most predictive of belonging to a chronic-like clade (Fig. 3a).

We noted that different variants were associated with different mutations. For example, S:E484K that we originally observed in many WT chronic infections, was associated with Alpha and Delta, but not with Omicron backgrounds, where an S:E484A had been fixed as a lineage-defining mutation. While only a few synonymous mutations were detected by LIME, interestingly, these were associated with many different background variants (Fig. 3b). Moreover, all shared spike receptor binding domain (RBD) mutations were associated with antibody evasion. Finally, we noted that a recent analysis based on global prevalence along transmission chains, has shown that a large proportion of mutations in SARS-CoV-2 are under purifying selection and are predicted to have low fitness[43]. On the other hand, more than two thirds of mutations with LIME score >0.05 were inferred as high-fitness mutations (Fig. 3c; fitness >0.1). When focusing on the low-fitness mutations, interestingly a few spike mutations came up, namely, S:S371F, S:G476D, S:V597I, S:R685H, S:T1136I (Supplementary Dataset 6). Moreover, two low-fitness mutations (ORF1a:T1638I, E:T30I)

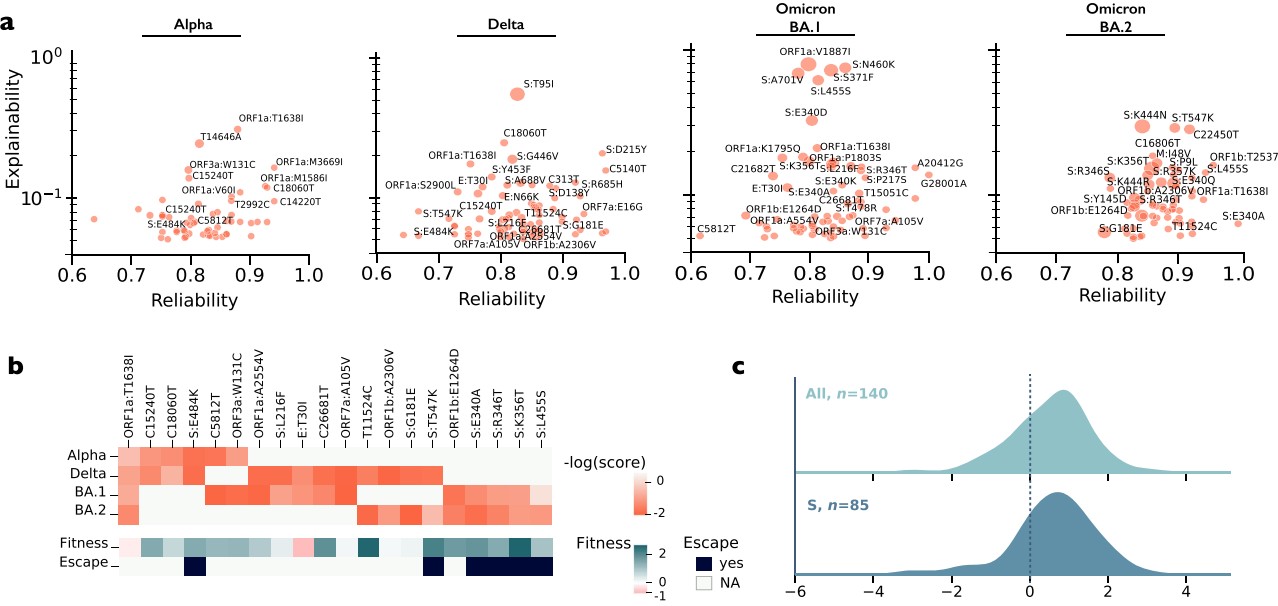

**Fig. 3 | Mutations associated with chronic-like clades. a** Mutations that best explain model predictions of chronic-like clades. Each panel depicts the mutations per a given background variant, with only those with an explainability score higher than 0.05 (75th quantile) shown (*y*-axis). The *x*-axis represents the average LIME R$^2$ score, indicating the prediction reliability. Dot size correlates with the number of samples in which the word was observed. For clarity only mutations with an explainability score higher than 0.1 are labelled, as well as those with a score higher than 0.05 that recur across different background variants (**b**). The full list of mutations is available in Supplementary Dataset 6. **b** Recurrent mutations associated with chronic-like clades, across different background variants. All mutations with an explainability score higher than 0.05 that appeared in at least two variants were used for the intersection. Fitness is based on inferences derived from mutations abundance across all globally circulating sequences[43] and antibody escape is based on inferences from deep-mutational scans using a variety of different types of antibodies[44,70]. **a, b** Synonymous nucleotide substitutions are denoted by their genomic position, whereas amino-acid replacements are denoted by the protein name and amino-acid replacement. **c** Distribution of fitness effect values for all *n* mutations with LIME scores higher than 0.05, shown for all genes in the genome (but, accessory genes under little selection were omitted) in the upper panel and for the spike gene in the lower panel. The dashed line represents fitness effect of zero (indicating neutrality).

were observed repeatedly against different variant backgrounds (Fig. 3b). We elaborate on these in the discussion.

Next, we tested whether mutations associated with chronic-like clades can predict successful mutations in future variants, i.e., variants that emerged after the dates when the chronic-like clade was sampled (see Fig. 4a). We focused on predictions derived from BA.1/BA.2 clades and on mutations in the spike RBD. As described above, this domain, embedded in S1, tends to be enriched for immune-evasion mutations and impacts ACE2 binding[44–47]. We used three measures to assess "success" of a mutation in global setting: (i) strong convergence along the phylogenetic tree, (ii) high global prevalence of the mutations, and as described above, (iii) high inferred fitness of mutation along transmission chains[43].

A set of 11 RBD mutations have been previously detected as undergoing rampant convergent evolution since the emergence of BA.2[48], most likely signifying they are adaptive mutations (Fig. 4b). Six of these eleven positions (55%) were inferred with LIME scores >0.05 in BA.1 or BA.2 chronic-like clades that preceded or coincided with the date when these mutations began increasing in frequency (Fig. S4).

We then performed the reverse analysis, which is essentially prediction in hindsight: we counted a total of fourteen RBD positions that bear mutations with LIME scores >0.05 in our BA.1/BA.2 chronic-like clades (Fig. 4c). Six of these fourteen mutations (>40%) go on to achieve a global frequency *f* > 30%, and nine mutations (64%) achieve *f* > 1% (Figs. 4a, S4). A partially overlapping list of ten of these fourteen RBD mutations (>70%) are inferred as high fitness mutations (fitness >0.1) in a global setup. Finally, six of these fourteen (>40%) mutations form part of the convergent mutations described above. All in all, these results suggest that chronic-like clades bear potential predictive value for successful RBD mutations (see discussion).

Our results suggest that chronic infections speed up the probability and hence the rate of adaptive evolution. Across balanced sets of equally sized clades, there were more non-synonymous mutations in chronic-like clades, particularly in the S1 domain of spike and in the RBD (Fig. 1d, Table S3). Accordingly, it is possible that increasing the viral population size in the community, which is akin to more transmission chains, would lead to an increased probability of observed beneficial mutations. To test this, we increased the sample size of controls and tested the fraction of control clades where we observe the list of convergent spike mutations in Fig. 4b. We found that a 10–20-fold increase in the size of the controls led to equal probabilities of observing these adaptive RBD mutations, likely associated with immune-evasion, between controls and chronic-like clades (Fig. 4c).

Finally, we went on use the model to predict chronic-like infections from sequences with missing metadata (see Fig. S7 for general statistics on these clades). To this end, we used the "words" (i.e., mutations) that we previously found as most strongly associated with controls or chronic-like clades and searched for clades with the highest probability of being derived from chronic infections. The number of predictions naturally depends on the prediction score cutoff we define (Fig. 5a). Given that we detected 271 chronic-like clades from 25% of the data that had reliable metadata, naively we expected three times more such clades (~800) in the remaining 75%. A prediction score cutoff between 0.6 and 0.7 resulted in such an estimate, and this was supported by an analysis of clades with partial metadata (*p* < 0.05, permutation test, Methods). We present two examples of clades inferred with a prediction score higher than 0.65 in Fig. 5b, c. One of these cases is actually a bona fide chronic infection where sequences were submitted with missing metadata. This case was identified by directly contacting the authors[49] (Fig. 5c; Methods), and the second is a completely novel prediction.

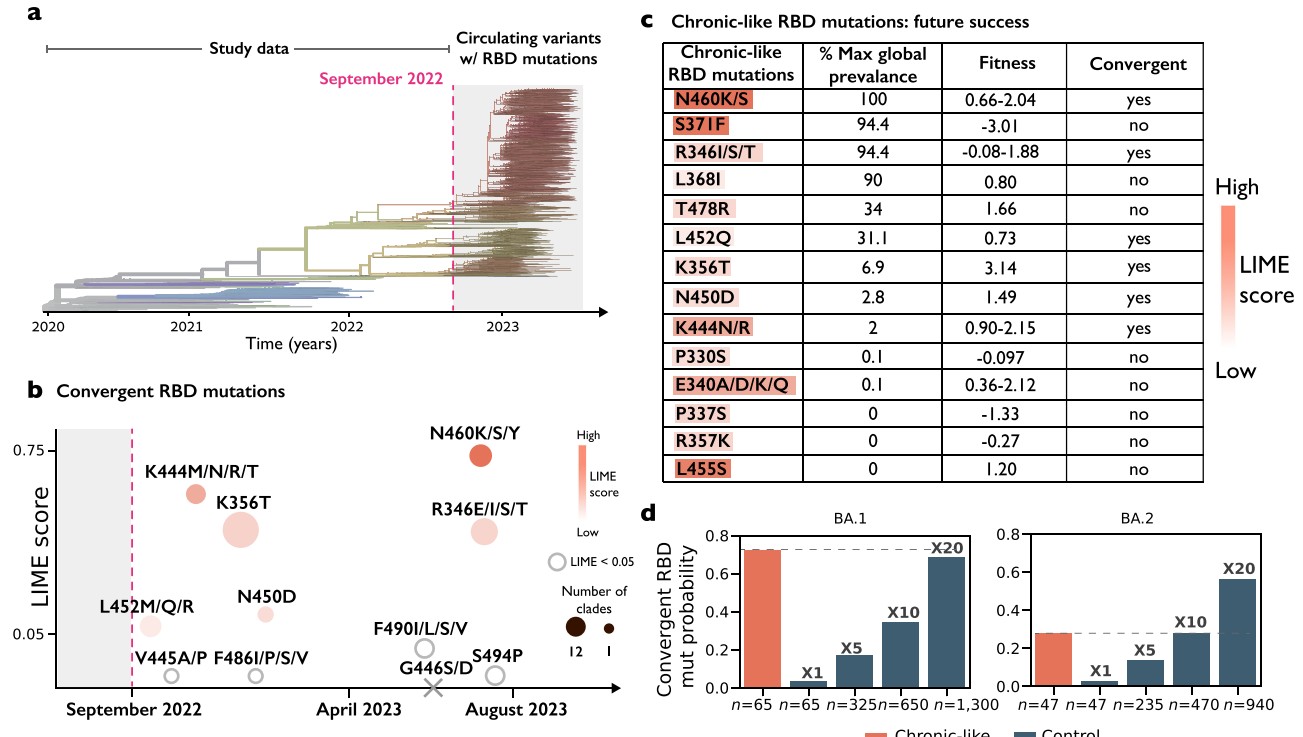

**c  Chronic-like RBD mutations: future success**

| Chronic-like RBD mutations | % Max global prevalance | Fitness | Convergent |
|---|---|---|---|
| N460K/S | 100 | 0.66-2.04 | yes |
| S371F | 94.4 | -3.01 | no |
| R346I/S/T | 94.4 | -0.08-1.88 | yes |
| L368I | 90 | 0.80 | no |
| T478R | 34 | 1.66 | no |
| L452Q | 31.1 | 0.73 | yes |
| K356T | 6.9 | 3.14 | yes |
| N450D | 2.8 | 1.49 | yes |
| K444N/R | 2 | 0.90-2.15 | yes |
| P330S | 0.1 | -0.097 | no |
| E340A/D/K/Q | 0.1 | 0.36-2.12 | no |
| P337S | 0 | -1.33 | no |
| R357K | 0 | -0.27 | no |
| L455S | 0 | 1.20 | no |

**Fig. 4 | Future adaptive RBD mutations forecasted by chronic-like clades.**
**a** Timeline of the study data indicating the last date of sampling (dashed) line, projected onto a phylogeny of SARS-CoV-2 derived from nextstrain.org[3].
**b** Convergent RBD mutations that occurred in sub-lineages of Omicron circulating since the summer or fall of 2022. Mutations are color-coded based on their maximal association with the group of chronic-like clades, with stronger shades of orange indicating a stronger association. X marks a mutation not detected. **c** All RBD mutations found in BA.1/BA.2 chronic like clades with LIME scores >0.05 are listed. The three right columns are measures of future global success of these mutations.

Max global prevalence refers to the maximal global proportion at which this mutation was observed, at any time point in the pandemic (Fig. S4), fitness effect range is based on inferences derived from mutation abundance across all globally circulating sequences[43], and convergence is described in the main text.
**b, c** When several mutations are listed (e.g., N460K/S) the color coding is based on the maximal LIME score obtained. **d** The probability of detecting RBD convergent mutations in chronic clades compared to their probability of detection in increasingly large sample sizes of control clades.

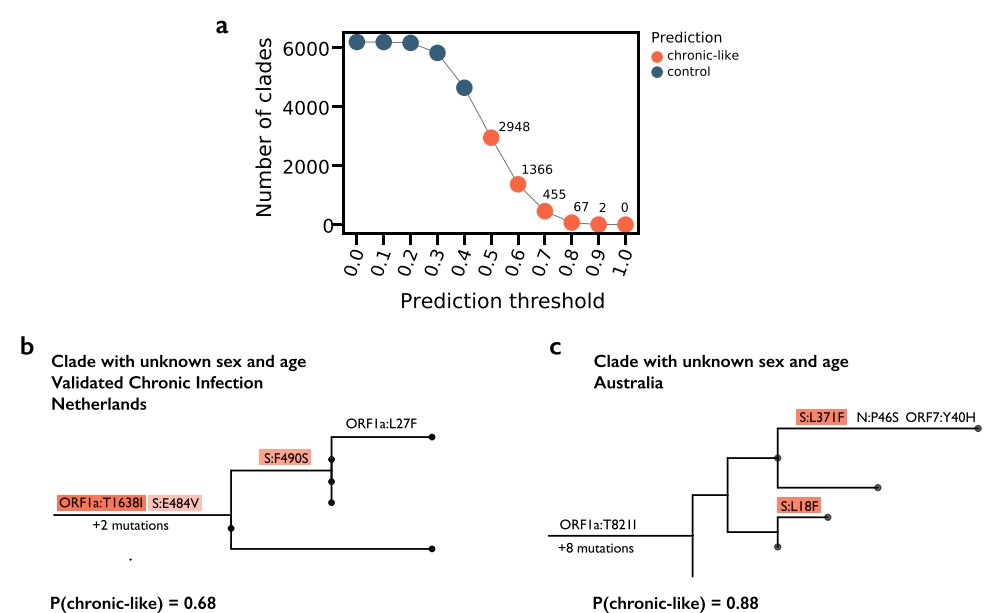

**Fig. 5 | Prediction of chronic-like infections in the absence of metadata and phylogeny. a** The number of chronic-like clades inferred given different prediction thresholds. **b, c** Two examples of clades inferred as chronic-like infections, with mutations driving the prediction color-coded as in Fig. 2.

## Discussion

In this work, we harnessed a very large volume of SARS-CoV-2 sequencing data and metadata to search for sequences that are likely derived from chronic infections. Our approach relies on the idea that there is deposition of viral sequences and metadata from de-identified individuals. In some cases, repeated sequencing is performed from the same individual who may be chronically infected. We search for signatures of such chronic infections by first directly mining a huge phylogenetic tree and next by using deep learning approaches that rely on our first findings. We go on to show that the chronic-like clades we find are enriched for high-fitness mutations, and that they allow to some extent prediction of RBD mutations that are later part of circulating strains.

There are some important limitations to the approach we put forth. First, there is no realistic way that we can verify each of our inferences of chronic-like clades. However, the analyses we show herein suggest that this set of clades is strongly enriched for chronic infections. Second, we most likely dramatically underestimate the number of chronic-like clades: we do not account for chronic infections sampled less than three times, and we do not account for chronic infections with low-quality sequencing or absent metadata. The lack of metadata in ~75% of sequences calls for the establishment of a more rigorous framework for standardized metadata submission. Moreover, we do not account for chronic infections that lead to onwards transmission. A recent paper detected onwards transmission in 3% of chronic infections[27]. This is probably a lower bound of detection but is consistent with the idea that there is low (but non-zero) probability for onwards transmission from chronic infections. This does however imply that we might be overlooking chronic infections with features that allow such onwards transmission to occur, which may be of particular interest. We tested whether we could detect onward transmission in our data (Methods). However, the structure of the tree (multiple polytomies) and lack of metadata make this exceptionally challenging, and thus future work is necessary to detect onwards transmission from chronic infections.

We interpret the results we found with regards to chronic-like clades. Interestingly, we found a lower rate of non-synonymous divergence in Delta as compared to BA.1. We could not conclusively pinpoint which gene or genomic region was responsible for this difference; we noted a borderline insignificant p-value for the spike gene ($p = 0.1$, ANOVA, $p = 0.06$, Tukey test; Table S3). Moreover, it was not clear whether the difference was driven by a higher rate of adaptive evolution in BA.1 chronic infections, possibly due to increased use of monoclonal antibodies, or due to an inherently lower rate of adaptive evolution in Delta chronic infections. The latter would be interesting, as Delta is the only VOC that lacks the N501Y mutation, which has been shown to form epistatic interactions that enable multiple additional antibody escape mutations[39–41]. Most of our Delta chronic-like clades were between May 2021 and March 2022, when vaccination and convalescence rates varied widely across the globe, and this may have impacted the probability of antibody escape mutations. However, we do not have enough data to shed more light on this finding.

We go on to discuss the features of the mutations most associated with chronic infections (Fig. 3a). The strongest signal was most often derived from non-synonymous spike mutations that have been shown to promote antibody evasion and possibly also ACE2 binding, but was also driven by mutations in other genes (Fig. 3b). Interestingly, some of the mutations that come up in chronic-like clades are very low fitness mutations, and we suggest this may represent either different selection pressures and/or epistatic interactions that emerge during chronic infections. For example, the low-fitness S:G476D is always accompanied in our chronic-like clades by S:G446V, on a Delta background, suggestive of epistasis. On the other hand, the two low fitness mutations E:T30I and S:R685H have been found during passaging of the virus in different types of cells[26,50,51], suggesting they confer an advantage in different tissues or under different conditions in the cell. Finally, ORF1a:T1638I (nsp3:T820I) is a low fitness mutation most commonly associated with all variant backgrounds in our chronic-like clades (Fig. 3b), and was found to be associated with HLA-A*01:01-allele restricted escape from CD8 + T-cells[52]. If so, it is possible that it may be advantageous on the background of certain HLA backgrounds but may be disadvantageous on other backgrounds, which could explain the absence of this mutation in the global phylogeny. Alternatively, it may have a different negative impact on transmission.

Our results highlight that adaptive evolution is enhanced during chronic infections as compared to evolution along transmission chains. This may be due to several non-mutually exclusive reasons: higher selective pressures, large viral population size, and the ability to cross fitness valleys through epistatic interactions among mutations[21]. Regardless of the reason, we show here that chronic-like clades hold some predictive power regarding mutations that later become globally successful. Of note, the correlation we find does not impose causality, i.e., we do not claim that the clades derived from chronic infections are direct predecessors of emerging variants. We further note that many successful mutations are missed out in chronic-like clades, and many predictions do not pan out. For example, mutations at positions S:E340 and S:P337 are rarely observed in global sequencing data, yet are strongly associated with escape from therapeutic monoclonal antibodies[53,54]. They may be less relevant in the context of a more full-blown immune response. All in all, this suggests that combining different approaches (e.g.,[30,44,55,56]) may allow obtaining a more accurate and precise predictive set of mutations.

The analysis of sequencing data often heavily relies on phylogenetic methods and models that are quite computationally expensive. As the volume of biological sequence data continues to increase (for example, over 4 million SARS-CoV-2 sequences were deposited during the last year), language models are emerging as valuable tools[30,57,58]. We have put forward the idea of using language models and deep learning to infer mutations associated with chronic infections, and further show as a proof-of-concept that the models can be used to infer chronic-like clades in the absence of metadata and even in the absence of a phylogeny. In an era of ever-growing pathogen sequencing, we envisage that monitoring chronic-like clades, in conjunction with other approaches, may be a valuable tool for predicting key driver mutations.

## Methods

### Sequence dataset curation and pre-processing

We accessed the GISAID database[59–61] on September 17th, 2022 and downloaded a total of 13,165,623 SARS-COV-2 sequences with their associated metadata. We began by employing stringent quality control and used an initial filtering step based on Nextclade version 2.5[62], which scores sequences based on a series of comprehensive criteria. We used all Nextclade defaults but changed the mixed sites threshold to 30. We also excluded from these criteria the "private mutations" criterion since we considered private mutations may very well characterize chronic infections. Sequences with a final quality score of "bad" or "mediocre" were removed from our analysis, while only those labeled as "good" were retained. Additionally, we removed sequences with ambiguous or conflicting dates (missing/partial dates, or a submission date earlier than collection date). We manually curated the age and sex metadata fields, resolving conflicts resulting from reporting in languages other than English. We considered ages below one year as one year old and maintained age ranges as provided (e.g., two samples labeled as age 10–19 were considered as samples of the same age). Samples where metadata was missing or corrupted were labeled with "unknown" in the relevant field. Overall, these steps resulted in a reduction of approximately 11% of the original number of sequences and yielded a final dataset of 11,717,404 sequences.

## Phylogeny-based inference of chronic-like clades

We used a global SARS-CoV-2 phylogenetic tree that is constantly updated with sequences added to the GISAID database. The tree was reconstructed using the UShER algorithm[63] and was kindly provided by Angie Hinrichs on August 25th, 2022. The mutational path of each sequences is also included with the UShER tree, and includes for each sequence the step-by-step mutations that occurred from the root of the tree till each leaf, based on ancestral sequence reconstruction[63]. We used this tree to identify "chronic-like" clades, defined as clades potentially derived from an individual with a chronic SARS-COV-2 infection:

Formally, we define a "chronic-like" group $M$ in the tree $T$ as a group of sequences $\{s_1, \ldots, s_n\} \in T$ that meets the following criteria: $M$ defines a monophyletic clade with at least $n = 3$ leaves but no more than 40 leaves; all $s_i \in M$ share the precise same location, the sequences $\{s_1, \ldots, s_n\}$ span at least 21 days, and 75% of the sequences in $M$ share the same age and sex (excluding "unknown" samples). Notably, we relaxed the last assumption from 100% to 75%, after extensive manual testing that revealed that in many cases sporadic sequences were included in the clade either erroneously or correctly but with missing metadata. We recovered sequences with ambiguous dates that were excluded in the initial quality control if they were part of a candidate clade (e.g., a sequence excluded since it was reported as February 2021 and was re-included if the chronic-like clade spanned this month).

Given the high proportion of data with an "unknown" label, we extracted additional clades to be further tested if they are "chronic-like", using our LM described below. We define an unknown group $U$ in the tree $T$ in the same manner as defined above, except that both age and sex are unknown. This yielded 18,760 clades.

## Control clades

Finally, we define a control group $C$ in the tree $T$ as a group $C = \{s_1, \ldots, s_k\} \in T$ that meets the following criteria: $3 \leq k \leq 40$, all $s_i \in C$ share the same location, the sequences $\{s_1, \ldots, s_k\}$ span at least 21 days, yet we ensured that the ages and sexes differ, i.e., at least four different combinations of age and sex exist in the group. This yielded 15,163 clades.

Due to the large difference in sample size between the cases and controls, we used bootstrapping to control for sample size. Specifically, we performed stratified sampling with replacement of $n = 10^4$ subgroups from the control group, each sized 271 (identical to the number of chronic-like clades), stratified by the Nextclade clade to allow for similar background variant distribution. This approach allowed us to calculate average values for the control clades across different measures described below.

## Bona fide chronic infections

We relied on our previous publication that includes $n = 27$ chronic infections[14,49] and added on sequencing data from $n = 5$ Omicron chronic infections[49] (data was obtained by directly contacting the authors).

## Assignment of mutations to clades

We set out to find the within-clade evolution of each of the chronic-like clades that we inferred. Notably, the UShER mutational path reports nucleotide mutations and lacks indels whereas the Nextclade annotation includes amino-acid replacements and indels. Therefore, for each clade (chronic-like or control), we first extracted the set of all mutations from the sequences using Nextclade mapping[62]. Then, we intersected these data with the UShER mutational path and removed from this set all mutations that occurred up to the ancestral node of each chronic-like clade. We included the mutations on the branch leading to this. At this stage we also excluded indels from this analysis since they were not included in the UShER mutational paths. Of note, all mutations in this manuscript are reported with respect to the ancestral Wuhan-Hu-1 reference genome sequence (GenBank ID NC_045512).

## Position masking

Similar to previous work[14,35,43,64], we masked all lineage defining mutations from the analysis since we noted that some sequences were erroneously assigned with the reference sequence nucleotide, presumably when sequencing coverage was low and bioinformatics pipelines made automatic erroneous assignments. To this end each sequence was assigned a clade based on Nextclade, and lineage-defining mutations were based on https://github.com/neherlab/SC2_variant_rates/blob/master/data/clade_gts.json[35]. Additionally, we masked mutations flagged as problematic positions in this table (https://github.com/W-L/ProblematicSites_SARS-CoV2/blob/master/problematic_sites_sarsCov2.vcf). Finally, we masked the two positions upstream and downstream of each masked position described above.

## Binned distributions

To compare the distribution of mutational counts between the chronic-like clades and controls, we divided the entire genome into 500-position long bins, denoted as $bin_i$, where $bin_i$ includes all mutations observed in positions $[i \cdot 500, i \cdot 500 + 500)$. We used a binomial test to identify bins significantly enriched for mutations.

## Sackin index

We extracted the sub-tree for each clade using the UShER platform[63], and calculated the Sackin index[65] as a measure of tree imbalance using Python's dendroPy framework[66]. Other indices were assessed (e.g., B1, Treeness) and deemed inappropriate for the task at hand.

## Entropy-based tree imbalance metric

We quantified tree balance using entropy of the node distribution across the hierarchy within a tree. The entropy calculation involves analyzing the distribution of nodes among tree levels, with lower entropy values indicating greater balance and higher values suggesting increased imbalance. Formally, the proportion of nodes in each level is given by $p_i = \{\frac{n}{N} \mid n \in T_i\}$, where $N$ is the total number of nodes in tree $T$, and $T_i$ is the i-th level of $T$. For a tree $T$ we obtain the normalized Shannon entropy $H(T) = -\sum_i p_i \log p_i$, by $\frac{H(T)}{\log N}$.

## Onward transmission from chronic-like clades

We used the following approximation to assess the number of potential transmission chains originated by chronic infected individuals. Given the 271 chronic-like clades we look for the ancestor of the clade in the tree and examine whether a direct descendant of the ancestor shares the same metadata as the chronic-like clade inspected. This will be served as the candidate set. More formally we define $T_C$ as the subtree containing only clade $C$ and $T_C^i$ as the subtree containing the ancestors up to level $i$, such that, $T_C^1$ will mark the subtree containing clade $C$, its ancestor and all of its descendant etc. $T_C^i$ will be considered as a potential onward transmission if a sequence with the same metadata (sex, age, location) is found in $\{s \mid s \in T_C^i, s \notin T_C, |T_C^i| < 5000\}$ and for each clade we select $\min_{i \leq 3} T_C^i$, $T_C^i$ is transmitted. We define the chronic-like clade nearest neighbors as all the sequences in the selected subtree $\{s \mid s \in T_C^i, s \notin T_C, |T_C^i| < 5000\}$.

## Linear regression for within clade mutation accumulation rate

We used an ordinary least-square (OLS) linear regression model to assess the slope of each chronic-like and control clade. For a given clade, we obtained for each sequence the number of within-clade mutations and regressed the number of mutations per sequence against date. OLS was performed using the python statsmodels version 0.13.5[67].

## Language model for mutation representation

To create a corpus of all mutations in the data, we used the set of sequences described above and constructed "sentences" comprised of "words" (tokens), each representing a mutation relative to the reference sequence. We included only mutations in coding regions and used the mutation annotation by Nextclade. Non-synonymous mutations were represented by the gene where they occurred and the associated amino acid replacement (e.g., S:D614G), while synonymous mutations were represented by the genome location and associated nucleotide change (e.g., C3067T). Deletions and insertions were also included and were represented as obtained by NextClade (e.g., 27871 for a deletion, 20:GGA for an insertion)[62]. For each sample in the GISAID dataset, a sentence was constructed to encompass all the mutations present in that sample. These mutations were sorted based on their genomic location to ensure a coherent representation within the sentence.

We limited the model vocabulary to mutations that appeared at least 45 times in the dataset, to allow for a reasonable vocabulary size of 38,000 unique tokens. Next, we trained a BERT model[38] from scratch on the task of masked language modeling to generate a numerical representation for each sample in the dataset. We excluded sequences that were selected as chronic-like clades, control clades, and clades with unknown metadata that were used later for classification and prediction. This resulted in a dataset of 10,646,407 sequences. We used 90% of the data for training and the remaining 10% for validation.

For tokenization, we applied a custom BERT tokenizer from the Hugging Face library[68] that splits sentences into tokens based on whitespace. We set the sentence length limits to be between 5 and 160 words. During training, the masking probability was set to 0.2, with a per-device training batch size of 10 and a validation batch size of 64. We used gradient accumulation steps of 8, which resulted in the final training batch size of 640 and a validation batch size of 512. The model was trained with the default optimizer AdamW[69] for two epochs with an initial learning rate of 1e-5. The best model was chosen based on the minimal evaluation set loss.

The model was trained on a single NVIDIA RTX A6000 GPU with 48 G RAM and 8 CPUs, taking a total of 48 h.

## Chronic-like clade classification

We used our pre-trained BERT model to classify chronic-like clades versus controls. Each clade was represented by a sentence that included all of the clade's corresponding mutations. Specifically, clade $c_i$ was represented by the sentence $s_i = \{"m_1 \, m_2 \ldots m_n" \, | \, m_j \in S(c_i)\}$, where $m_j$ is a mutation (token) and $S(c_i)$ denotes all sequences within clade $c_i$. To ensure consistency, mutations were sorted based on their genomic location.

To prevent potential confusion between lineage-defining mutations that occurred in the past and their subsequent appearance in chronic clades, all lineage-defining mutations according to the Nextclade variant were removed. Masking of problematic positions/mutations was performed as described above.

The classification process was performed based on VOC chronology, utilizing three distinct folds. The dataset was divided into five main groups: pre-VOC, Alpha, Delta, BA.1 and BA.2, along with the remaining variants. For the non-pre-VOC variants, we considered the preceding variants for training and tested on the relevant variant (Table S1).

To ensure a balanced representation of the control data, a down-sampling technique was applied. Initially, the control clades were filtered based on the Sackin index, including only those with a value lower than 1.44 (corresponding to the average of control clades inferred herein, Fig. 1). The objective was to intentionally select control clades that are most likely derived from transmission chains and unlikely to have derived from a chronic infection. Following the initial

sampling, we performed an additional round of down-sampling down to $n = {\sim}270$ control clades, to ensure that the Nextstrain background variants and clade size are balanced across the set of controls and chronic-like clades.

To optimize the classification process given the relatively small sample size, we modified the BERT for Classification architecture obtained from Hugging Face. Specifically, we froze all embedding and encoder layers, allowing changes only to the last layers responsible for pooling and classification. The training procedure encompassed 30 epochs, and for each fold, the model with the lower evaluation loss was selected (Fig. S5). Throughout the training phase, a batch size of 64 was employed, while a batch size of 32 was used for the evaluation set. To assess the performance of each model, we utilized two metrics: ROC AUC and AUPR (weighted by clade size and the number of mutations).

## Classifier performance assessment using partial metadata clades

We focused on clades with partial metadata, specifically those with either the same age and 75–100% unknown sex or the same sex and 75–100% unknown age (total of 79 clades). Additionally, we included control clades (total of 5305 clades), which were excluded during the original classifier training and predictions. Focusing on the Omicron fold, which offered the most extensive data, we utilized a trained classifier to estimate the probability of a clade being chronic-like. To investigate whether clades with partial metadata are more likely to be classified as chronic-like, considering they exhibit greater meta-data agreement, we conducted a permutation test on the percentage of clades surpassing a specified threshold. Given the limited sample size of the partial metadata group, we generated the background distribution using the control clades. Specifically, we randomly sampled $n = 79$ clades while controlling for clade size, repeating this process 10,000 times. Subsequently, we computed the empirical one-sided $p$-value to determine the likelihood of observing the actual proportion of partial metadata clades compared to the control background distribution. The permutation test was conducted using a 0.6 threshold, as the number of predictions exceeding thresholds of 0.7 and higher was very small.

## Model explainability

Local Interpretable Model-Agnostic Explanations (LIME)[42] was applied to understand the underlying reasoning behind how the classifier inferred chronic-like clade. In each test fold (Alpha, Delta, BA.1, BA.2), we identified the top 10 mutations with the highest LIME scores for each clade. These scores were aggregated by introducing a mutation score per background variant, denoted as $S_v(m)$, which sums the LIME scores across all samples within variant $v$'s test set. This aggregation approach enhances the significance of mutations that consistently appear across multiple samples, reinforcing their predictive potential. Additionally, as LIME fits a regression line per clade in order to generate inferences, the $R^2$ serves as a measure of result reliability, and here it was was averaged across clades.

## Reporting summary

Further information on research design is available in the Nature Portfolio Reporting Summary linked to this article.

## Data availability

All data supporting the findings of this study have been deposited in the Zenodo database under accession code https://doi.org/10.5281/zenodo.10338988 (https://doi.org/10.5281/zenodo.10338988). These include the 13,165,623 sequences and metadata originally obtained from GISAID on September 17, 2022, EPI_SET_230725to (https://doi.org/10.55876/gis8.230725to), and available on Zenodo as supplementary dataset 7. The data of this study further include the chronic-like

**Article** https://doi.org/10.1038/s41467-024-44803-4

and control clades identifiers, mutations, and sub-trees derived from the global phylogeny. They also include language model raw corpus files, trained models, and all predictive mutations for the chronic-like clades groups. The full description is available in the Supplementary information and in the Readme file on Zenodo.

## Code availability

All models used for training and scripts used for analysis are available in the repository https://github.com/Stern-Lab/chronic-covid-mlm. The paper code release is also available in the Zenodo database under accession code https://doi.org/10.5281/zenodo.10339153 (https://doi.org/10.5281/zenodo.10339153).

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

## Acknowledgements

We would like to thank Pleuni Pennings, Jesse D. Bloom, Avigdor Eldar, Daniel Weissman and Angie S. Hinrichs for helpful discussions and comments on the manuscript. We gratefully acknowledge all data contributors, i.e., the Authors and their Originating laboratories responsible for obtaining the specimens, and their Submitting laboratories for generating the genetic sequence and metadata and sharing via the GISAID Initiative, on which this research is based. This study was supported by grants to AS: an ERC starting grant 852223 (RNAVirFitness), an Israeli Science Foundation grant 1930/22, and a grant from the Tel Aviv University Center for Combatting Pandemics. This study was also supported by fellowships to S.H. and D.M. from the Edmond J. Safra Center for Bioinformatics at Tel Aviv University and a Fellowship to D.M. from the Azrieli Foundation.

## Author contributions

Conceptualization: S.F., A.S., D.M. and S.H. Phylogenetics and bioinformatics: S.H. and D.M. Initial proof of concept analyses: S.F. Language model design and implementation: D.M. and D.B. Supervision: D.B and A.S. Writing: S.H., D.M, A.S. All authors read and revised the paper.

## Competing interests

The authors declare no competing interests.
