## [Peer Review File · Nature Communications]

Using big sequencing data to identify chronic SARS-Coronavirus-2 infectionsEditorial Note: This manuscript has been previously reviewed at another journal that is not operating a transparent peer review scheme. This document only contains reviewer comments and rebuttal letters for versions considered at *Nature Communications* .

REVIEWERS' COMMENTS

Reviewer #1 (Remarks to the Author):

I thank the authors for the comprehensive review.

Reviewer #2 (Remarks to the Author):

The authors have done various analyses to address comments from the previous version. I have a few remaining comments:

1. I'm surprised (and pleased) that 4.3M more sequences have been deposited since late 2022. I appreciate the authors correcting my misapprehension about the rate of sequencing last year. While I hesitate to ask that everything be done again, it does naturally raise the question of whether key findings can be replicated by looking at that extra data? e.g. some of the key observations about mutation pattern?

2. I am unconvinced by the early pandemic comparison. Zero of 27K is significant, but the interpretation is very sensitive to small counts. For example, just 1 or 2 "hits" in the early data would be consistent with a rate of half of the observed events in the main dataset being something other than chronic infections, and 5 or 6 would mean there is no significant difference. Furthermore, there are many confounders between the sampling in early and mid/late pandemic, which makes it hard to know what is driving the apparent difference. The authors point out one possible explanation, but it's not dispositive. I also don't think the nearby branch analysis is that informative. What do we really know about the distribution of sources from next-nearest clades (e.g. whether they are truly close in time/space or not?) I think on average it is hard to predict.

To be clear: both of these ideas were reasonable, I just think that when mining these public datasets with essentially no idea how they were sampled one has to be very explicit about the limits and possible sources of sampling bias.

3. I don't think my comment about Figure 1D was addressed, perhaps because I was unclear. I'd like to see the Y axis as a rate (mutations in this bin per genome in this category), rather than a count (mutations in this bin in all the genomes in this category). I think it is highly informative to look at the rate of mutations (both genome-wide and per bin) as well as the regions where the rate is highest.

Reviewer #3 (Remarks to the Author):

The authors have satisfactorily addressed all my comments. I would like to recommend the manuscript for publication.

REVIEWERS' COMMENTS

Reviewer #1 (Remarks to the Author):

I thank the authors for the comprehensive review.

Reviewer #2 (Remarks to the Author):

The authors have done various analyses to address comments from the previous version. I have a few remaining comments:

1. I'm surprised (and pleased) that 4.3M more sequences have been deposited since late 2022. I appreciate the authors correcting my misapprehension about the rate of sequencing last year. While I hesitate to ask that everything be done again, it does naturally raise the question of whether key findings can be replicated by looking at that extra data? e.g. some of the key observations about mutation pattern?

Unfortunately, this requires re-doing the entire end-to-end analysis and modelling, which makes this quite time-consuming. To make it more clear to the readers that the data are constantly on the rise, we have added the following sentence to the discussion:

“As the volume of biological sequence data continues to increase (for example, over 4 million SARS-CoV-2 sequences were deposited during the last year)...”

2. I am unconvinced by the early pandemic comparison. Zero of 27K is significant, but the interpretation is very sensitive to small counts. For example, just 1 or 2 "hits" in the early data would be consistent with a rate of half of the observed events in the main dataset being something other than chronic infections, and 5 or 6 would mean there is no significant difference. Furthermore, there are many confounders between the sampling in early and mid/late pandemic, which makes it hard to know what is driving the apparent difference. The authors point out one possible explanation, but it's not dispositive. I also don't think the nearby branch analysis is that informative.

In line with the reviewer's suggestion, we have added a disclaimer to this result on page 5 of the Results:

“Of note, this is a small difference, and sampling strategies varied between early and mid/late stages of the pandemic, suggesting caution when interpreting this result.”

What do we really know about the distribution of sources from next-nearest clades (e.g. whether they are truly close in time/space or not?) I think on average it is hard to predict. To be clear: both of these ideas were reasonable, I just think that when mining these public datasets with essentially no idea how they were sampled one has to be very explicit about the limits and possible sources of sampling bias.

Indeed, in the text we emphasize that this is only a sanity check, and we use careful language on page 5 in accordance (bold added for emphasis):

“A second **sanity check** that we performed, was to examine sequences in the clades that neighbor our chronic-like clades. If chronic-like clades were mostly derived from an outbreak in a high-age setting, then neighboring clades would **probably** display higher ages as well. However, our results show that neighboring clades show the same average age as controls, which is lower than that of chronic-like clades (Fig. S2). Overall, this **suggests** that our chronic-like clades are enriched for samples from the same chronically infected individual.”

3. I don't think my comment about Figure 1D was addressed, perhaps because I was unclear. I'd like to see the Y axis as a rate (mutations in this bin per genome in this category), rather than a count (mutations in this bin in all the genomes in this category). I think it is highly informative to look at the rate of mutations (both genome-wide and per bin) as well as the regions where the rate is highest.

We have now added the requested figure as Figure S2c. This allows the reader to see the raw data (counts) as the main figure, and the normalized data (per genomes) as Figure S2c, with a more in-depth explanation in the supplementary text. This is written on page 6:

“Chronic-like and *bona fide* share three of four enriched S1 bins, whereas controls share only one enriched S1 bin with the other two categories; the average number of S1 mutations per clade was significantly higher in *bona fide* and chronic-like clades than in control clades ($p < 5 \cdot 10^{-3}$, Mann-Whitney U test; Supplementary text, Fig. S2).”

Reviewer #3 (Remarks to the Author):

The authors have satisfactorily addressed all my comments. I would like to recommend the manuscript for publication.